# Adaptive Memory Module for Sequential Planning and Reasoning

## Abstract

Efficient planning and reasoning in sequential decision-making tasks remains a core challenge for machine learning models. These tasks often involve intricate decision sequences leading to combinatorial complexity that hampers traditional planning methods. Humans on the other hand leverage flexible planning strategies and adapt their thinking time based on the complexity of the problem at hand to efficiently solve complex reasoning problems. Inspired by this, we propose and investigate an end-to-end memory-based adaptive learning algorithm to enhance planning capabilities and resource allocation of AI agents. Our study borrows concepts from adaptive computation and incorporates memory and reusability mechanisms into agents. This allows agents to meta-learn flexible reasoning strategies, plan deeper, and efficiently adjust their computation to not only improve inference time efficiency but also generalize to more complex problems. Finally, our study of the adaptive memory module reveals patterns comparable to human decision-making mechanisms such as increasing certainty, reconsideration and alternative exploration. This work contributes to the evolving understanding of harnessing adaptive computation to enhance machine learning models' capabilities in complex reasoning and sequential decision-making tasks.

## 1 Introduction

Planning and reasoning are integral to human decision-making and have yet to be adequately incorporated into AI models. These components are required in building highly capable general AI agents that can assist in scientific discovery and push the frontiers of human knowledge. Furthermore, reasoning in sequential decision making domains is characterized by an interaction between the past and the future, where information and decisions from previous steps significantly inform the present decisions while these decisions, in turn, influence future inputs and observations.

Recent trends (Wei et al., 2022; Yao et al., 2023; Nye et al., 2021; Zelikman et al., 2022; Lanchantin et al., 2023; Hu & Clune, 2023; Lewkowycz et al., 2022) have shown how simply scaling up model size, and dataset size can result in emergent capabilities like reasoning. However, it has been shown (Sawada et al., 2023) that these large language models are still weak at more complex reasoning tasks. Factors contributing to these limitations include low-quality step-by-step reasoning data available on the internet and difficulty in collecting high-quality reasoning data for complex reasoning problems. Collecting such data becomes very difficult for problems that humans themselves cannot solve and for visual reasoning problems. Another potential cause is the limited thinking time (Bubeck et al., 2023) before outputting the first token.

Another common approach taken towards building AI agents capable of planning has been studied under the umbrella of model-based reinforcement learning. In these approaches, the decision-maker constructs mental simulations of potential future states, actions, or outcomes within a decision tree, using a learned transition dynamics model. These methods (Schrittwieser et al., 2020; Silver et al., 2016; Pierrot et al., 2019; 2020; Hafner et al., 2023) rely on learning world models, and leveraging explicit search strategies like Monte Carlo Tree Search to build AI agents that can solve long horizon planning problems. These works also leverage additional decision/inference time to achieve superhuman level performance in various complex planning games like Chess and Go. However, they lack flexibility and adaptability and are often difficult to scale in more complex environments. Human on the other hand adapt their planning strategies and planning time based on the complexity of the

problem and urgency around the need to arrive at a decision. Works like Edwards et al. (2018), show the benefit of other planning routines involving hierarchical planning and reverse directional planning routines.

Works like Racanière et al. (2017); Pascanu et al. (2017); Guez et al. (2018); Gowal et al. (2018) have taken a step toward flexible and adaptable planning routines by allowing agents to learn to plan. However, these still rely on having an explicit transition dynamics model. Recently Guez et al. (2019) proposed to learn an end-end agent that meta-learns flexible planning routines via its memory. This helps agents learn and adapt their planning strategy, and avoid tree search based planning algorithms. Another recent work (Bansal et al., 2022), also shows similar effects in other planning domains. However, the computation time in the above approaches is fixed. These works have also shown how leveraging more computation at inference time for more difficult problems (Baldock et al., 2021), can help improve the performance of the model. Other works (Guez et al., 2019; Bansal et al., 2022; Jones, 2021; Hamrick et al., 2021) have also shown how additional computation time can help improve performance in complex planning domains. All these works however fix the computation time and do not allow the agent to adapt the number of repetitions taken by the memory module *e.g.,* LSTM with a fixed number of iterations.

The study of decision-making and planning extends to human psychology, particularly in scenarios where task complexity requires varying degrees of cognitive effort (Payne et al., 1988). Throughout our development, our engagement with the environment undergoes a series of changes, shaping our ability to navigate challenges and solve tasks. Early on, basic functions emerge in response to simple, repetitive tasks, forming the foundation for later complexities. As we mature, higher-order cognitive processes emerge in the brain, allowing us to handle more intricate situations and integrate conflicting information through complex deliberation. This progressive development aligns with the evolving demands of our surroundings, reflecting the adaptable nature of human cognition. Furthermore, it is clear from empirical data that human reaction times vary considerably based on task complexity (Payne et al., 1993; Bläsing & Bornewasser, 2021), where results have been shown that correct response times to a stimulus increase either with a greater number of stimuli or when there are many possible planning alternatives to be considered. For example, increased fixation time on a set of increasingly more complex, or contradictory, instructions to complete a task was observed using eye tracking (Ullman, 1984). There is a clear pattern relating task complexity in the form of information processing load and possible sequence of actions or interactions to complete a task. Notably, previous research has indicated that temporal pressure and the risk of critical irreversibility can hinder decision-making, even in situations demanding swift actions (Klein, 1999; Pahlke et al., 2011; Payne, 1976). Human decision-making exhibits a remarkable flexibility and responsiveness to environmental cues, often prompted by a sense of urgency (Reddi & Carpenter, 2000). These influences of urgency, whether biologically driven by factors like aging or environmentally-induced pressures, create a natural push for more efficient decision-making mechanisms, such as reflexes. As models scale up, the incorporation of such mechanisms becomes increasingly pertinent.

We extend prior work to suit the requirements of sequential decision-making scenarios, thereby enabling us to probe the effects of adaptive computation on the planning process. Earlier research by Banino et al. (2021); Graves (2016) has not only demonstrated the performance benefits of adaptive computation but has also highlighted instances where higher computation is needed, indicating increased task complexity. Understanding the role of these features in sequential reasoning tasks grants us insight into the underlying mechanisms of these models.

In this study we aim to explore how agents can leverage adaptive computation, memory, and an end-to-end learning framework to meta-learn flexible planning strategies and enhance decision time efficiency while retaining the ability for further deliberation to solve more complex problems.

We make the following contributions in this work:

1. **Analysis of Memory Modules**: Our investigation centers on agents equipped with single and multiple-step memory modules in the realm of sequential decision-making. This comprehensive analysis sheds light on the influence of memory integration on agents' planning efficiency and their capacity to generalize to more complex reasoning problems.

2. **Demonstration of Adaptive Computation within sequential decision making domains**: We present a compelling demonstration of the adaptive computation capabilities in our proposed architecture. By applying it to intricate puzzle-solving scenarios, we seek to

highlight the potential of our approach to amplify planning efficiency while concurrently upholding performance standards.

3. **Exploration of Complexity-Pondering Relationship**: Our research studies the interplay between information complexity within a state and the subsequent pondering steps carried out by an adaptive agent. We observe three distinct learned behaviors, namely increasing certainty, exploring alternatives, and reconsideration. These behaviors are interpreted through a psychological lens by analyzing the evolution of internal transformations within a trained agent. Our central hypothesis posits that the adaptive agent strategically allocates computation resources based on the complexity of input data, as well as the remaining difficulty of the task.

4. **Sokoban Expert trajectories Data**: We open source all the expert Sokoban trajectories collected for the experiments in this study, allowing others to further pursue research related to reasoning and planning.

## 2    METHODOLOGY

We investigate how memory, adaptive computation, and an end-to-end learning approach can be used to build an agent that can meta-learn flexible reasoning strategies without labelled reasoning data, reuse information across time steps and adapt its computation time based on input complexity to not only generalize to more complex reasoning problems but also efficiently solve easier tasks by halting earlier. We now introduce our agent and its memory module *i.e. Adaptive Memory Module* (AMM).

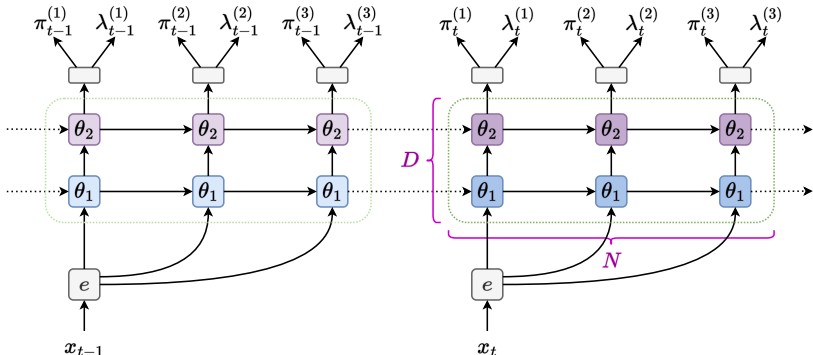

Figure 1: Adaptive Memory Module : The embedded input is combined with an initialized core state $h_0$, which is updated with the same input at each iteration, until the halting probability ($\lambda$) forces termination. The outcome of the memory rollout is then passed to a linear layer which outputs the action logits.

The input observation is first encoded by a CNN encoder. This three dimensional input is then processed by a recurrent memory module. We use the DRC architecture for memory (A.4) that has shown strong performance (Guez et al., 2019) in the environments investigated in this study. However, a diverse range of architectures ranging from Long Short-Term Memory (LSTM) (Hochreiter & Schmidhuber, 1997) and Multi-Layer Perceptron (MLP) (Rumelhart et al., 1986) to Gated Recurrent Unit (GRU) Cho et al. (2014) and Attention-based layers Vaswani et al. (2017), can be used, thereby imparting flexibility and adaptability to the model. The hidden states are then passed through the policy head to predict the action probabilities and the halting head to predict $\lambda$ which is used to decide when to halt. The internal memory is not reset after each step in the environment, enabling the agent to reuse past planning information. This allows the agent to plan deeper into the future and generalize to more complex reasoning problems.

In this study we use the PonderNet Algorithm (Banino et al., 2021) to perform adaptive computation and adapt it for the sequential decision making setting. This allows the agent to alter the number of computational steps it undertakes based on input conditions and it's internal memory that captures planning and reasoning information from the past. The core operation at each timestep is encapsulated in a step function of the form:

$$\hat{y}_t^n, h_t^{n+1}, \lambda_t^n = s(x_t, h_t^n),$$

where the symbol $x_t$ denotes the input at time step t, while $h_t^n$ represents the hidden state at time step t and iteration n, and $\hat{y}_t^n$ characterizes the predictive output at the n-th computational step. It is important to note that $x_t$ is the same input for each iteration of the step function, as can be seen in Figure1. The parameter $\lambda_t^n$ encapsulates the conditional probability of halting or terminating the ongoing process at the n-th computation step, conditioned that it has not halted previously, when making a decision at timestep t. The function $s$ can be instantiated as a diverse array of neural network architectures.

The unconditioned probability associated with the act of halting at the n-th step, denoted as $p_t^n$, is derived as follows:

$$p_t^n = \lambda_t^n \prod_{j=1}^{n-1}(1 - \lambda_t^j).$$

During training, predictions are gathered from each iteration, and individual losses for each iteration are calculated. Subsequently, a weighted averaging strategy is used to consolidate these losses, with the weighting scheme derived from the iteration-specific halting probabilities ($p_n$). Simultaneously, the step function is bounded by a predetermined upper limit on computational steps, as characterized by the variable $N$. The aggregate loss function for AMM, denoted as $L$, is composed as follows:

$$L = L_t^{Rec} + \beta L_t^{Reg} \tag{1}$$

$$L_t^{Rec} = \sum_{n=1}^{N} p_t^n \mathcal{L}(y_t, \hat{y}_t^n) \tag{2}$$

$$L_t^{Reg} = KL\left(p_t^n \| p_G(\lambda_p)\right) \tag{3}$$

This aggregate loss is the summation of two constituent components: the recurrent segment of the loss and the regularization term in the form of Kullback–Leibler divergence. $L_{Rec}$ captures the aggregate effect of losses across pondering steps, with $\mathcal{L}$ representing the loss function quantifying the dissimilarity between the target value $y$ and the predictive output $\hat{y}_n$. The regularization loss, denoted as $L_{Reg}$, is the Kullback–Leibler divergence ($KL$) between the distribution of halting probabilities and $p_G$. $p_G$ is the prior distribution represented by the geometric distribution parameterized by the $\lambda_p$, which takes the following form:

$$Pr_{p_G(\lambda_p)}(X = k) = (1 - \lambda_p)^k \lambda_p.$$

The regularization loss biases the network towards $\frac{1}{\lambda_p}$ expected prior number of steps and incentivizes non-zero probabilities for all pondering steps. In essence, it encourages the network to explore different possibilities and options.

During inference, the halting process is implemented via either a stochastic sampling mechanism, where the decision to terminate is made by drawing from the Bernoulli distribution with parameter $\lambda_t^n$, or is decided based on a simple probability threshold *i.e.* halt if $\lambda_t^n > 0.5$. Finally, the predictive output $\hat{y}_t^n$ is obtained from the time step when the network decides to halt.

$$halt_t^n = \text{Bernoulli}(\lambda_t^n) \quad \text{or} \quad halt_t^n = \lambda_t^n > 0.5$$

## 3 EXPERIMENTS

### 3.1 ENVIRONMENTS AND DATA

We have chosen the Sokoban Puzzle environment since it necessitates a certain degree of planning to determine a sequence of actions that precludes future complications arising from a lack of foresight. This challenging puzzle is presented as a two dimensional 7x7 grid where an agent must push boxes over targets in under a fixed amount of steps (Racanière et al., 2017; Guez et al., 2019; Botea, 2002). When increasing the number of boxes, it becomes more important to become aware of potentially irreversible moves that could be avoided with better planning. The input is a one-hot encoded transformation of the entire grid of the puzzle *i.e.,* a 7x7 grid with 7 different elements (*e.g.,* walls, boxes, etc.) becomes a 7x7x7 boolean matrix. There are a total of 8 possible actions: move or push in the 4 possible directions, and 7 possible elements in the state: walls, spaces, targets for boxes, boxes on target, and boxes not on target.

We use the same methodology as Botea (2002) to generate levels which are then split into train and test sets. To create the expert trajectories, we utilize the deterministic solver A* (Hart et al., 1968). The train set consists of 2.1M steps of Sokoban-small-v0 environment that contains 2 boxes. The validation sets consist of 1000 levels each for Sokoban-small-v0 (2 boxes) and Sokoban-small-v1 (3 boxes).

### 3.2 TRAINING

We use the behavior cloning algorithm (Torabi et al., 2018), wherein the agent learns from expert trajectories by predicting subsequent actions in the sequence. Specifically, a cross-entropy loss function between the expert's actions and the model's predictions is used.

For the static model, we directly calculate the loss between the action at the final pondering step and the expert action. Whereas, for the adaptive model, we use the PonderNet Loss (Equation 1) and use the weighted average of the cross entropy loss between the actions at each pondering step and the expert actions.

For optimization, we use the AdamW optimizer with weight decay. Our learning rate undergoes a linear warm-up followed by a cosine annealing decay for the remaining training iterations. The model undergoes training for a single epoch only. The details of the hyperparameter values can be found in the appendix (Tables 3, 4).

### 3.3 EXPERIMENT 1: IMPACT OF REPETITIONS IN STATIC MEMORY MODULE

In this section, we study the impact of augmenting the number of repetitions within the static memory module on the sample efficiency and generalization capabilities of the agent by comparing the average number of levels solved across both training and validation datasets. The underlying hypothesis posits that an increased number of repetitions allows the agent to plan deeper and encode longer-term interactions which should improve its reasoning capabilities in more complex environments. While this augmentation may not prove advantageous for every input state, the expectation is that it would, at the very least, not yield inferior performance compared to an identical model using only a single repetition.

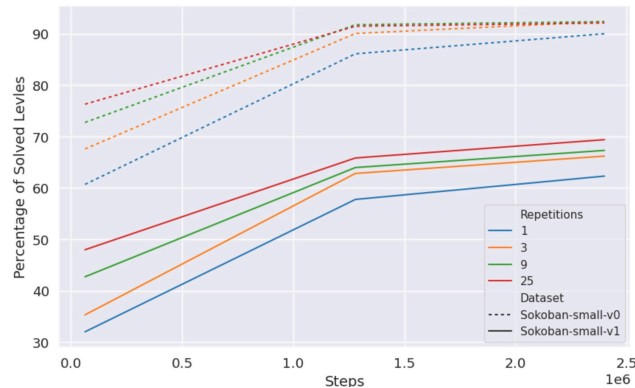

Figure 2: Mean fraction solve at different steps of training for various repetition counts. The results are mean results across three seeds.

Four models with increasing number of repetitions are trained on a subset of Sokoban-small-v0 levels and are validated on a different subset of Sokoban-small-v0 as well as Sokoban-small-v1 levels to test generalization to an environment with one more box.

We observe the following as presented in Figure 2:

**Sample Efficiency**: An increased number of repetitions indeed expedites the learning process. Models with more repetitions displays faster improvement, reaching comparable performance levels with fewer training steps, as evidenced by higher average fraction solved after a fixed number of steps.

**Performance on the same difficulty level**: We notice that as we increase the number of repetitions taken by the static memory module, they all converge to a similar performance at the end of training when evaluating the agents on puzzles with the same level of difficulty as the training set.

**Generalization to more difficult puzzles**: We observe a discernible enhancement in the fraction of puzzles solved by models with an increased repetition count, when evaluating on a more complex set of puzzles. This observed trend shows that increased repetitions contribute to improved planning and reasoning, particularly when addressing intricate decision sequences, helping improve generalization to more complex unseen problems.

### 3.4 EXPERIMENT 2: ADAPTIVE MEMORY MODULE

In this section we explore integrating adaptive computation into the memory module. We now allow the model to learn when and how much to ponder and compare the performance.

**Improvement in performance**: Here we explore whether the weighted and adaptive loss can help regularize and stabilize training for inputs of varying complexity instead of fixing the amount of computation performed for different inputs. The results in Figure 3 reveal that the adaptive planning model exhibited improved performance in general, when compared with its static counterpart, while only using 9 max repetitions during training. This observation holds true for both in-distribution (Sokoban-small-v0) and out-of-distribution (Sokoban-small-v1) settings. This suggests that the weighted and adaptive loss is helping improve the training process and the performance of the planning agent.

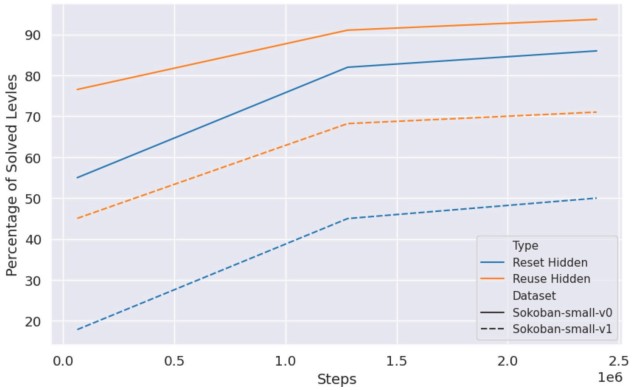

Figure 3: Performance and average number of repetitions used by Static vs Adaptive Memory module across validation datasets.

**Improvement in Generalization**: We hypothesise that inputs with higher complexity necessitate a greater amount of computational resources. We observe in Figure 3 that the agent is able to perform better on the more complex puzzles in Sokoban-small-v1 set compared to its static counterpart. We believe this is due to deeper planning by the agent by adapting and increasing its computation/number of repetitions as observed in Figure 3 and Table 1.

Moreover, we investigate the connection between early-episode pondering and the stage nearing level completion. Our analysis uncovers a statistically significant negative correlation ($r = -0.31, p = 6.14e^{-16}$) between the step number within an episode and the duration of pondering, when the agent chose to adapt it's computation (number of repetition smaller than the max). This indicates that, as the episode unfolds, the corresponding halting times exhibit a marginal reduction suggesting a decrease in the overall perceived complexity.

**Improvement in training and inference efficiency**: We can observe that the agent is able to adapt its computation time to have shorter inference time on easier inputs and increase the computation on more complex puzzles. This results in a lower average computation time measured by the mean number of repetitions taken

| Level | Mean | Std | Range |
|---|---|---|---|
| Sokoban-small-v0 | 6.76 | 2.66 | 8.00 |
| Sokoban-small-v1 | 7.14 | 2.31 | 8.00 |

Table 1: Pondering statistics across validation datasets. Shown is the result of taking the mean, standard deviation, and range of the number of ponder steps taken by a trained AMM over 1000 levels for each environment.

across the validation set. Results in Figure 3 and Table 1 show that the adaptive memory module takes far lesser number of average reptitions (6.76 for Sokoban-small-v0 and 7.14 for Sokoban-small-v1) compared to its static counterpart (25 repetitions) while still achieving better performance than the static counterpart. We also notice a large relative standard deviation, and a range that implies the AMM can and does halt at different steps, suggesting appropriate adaptation of pondering time based on necessity,

Additionally our best performing AMM agent only uses 9 repetitions during training which is far lesser than the best performing static agent that uses 25 repetitions, helping improving the training efficiency significantly.

## 3.5 EXPERIMENT 3: ROLE OF REUSING MEMORY IN SEQUENTIAL DECISION-MAKING DOMAINS

Here we explore if the adaptive memory module is able to reuse the memory state from past time steps to plan deeper in future time steps and improve generalization performance. As observed in Figure 4, we can see that the agent is indeed able to generalize much better when the hidden state is not reset, indicating reuse of planning/reasoning information from past time steps for deeper planning to predict better future actions.

## 4 VISUALIZING PONDERING STEPS

Here we create visualizations to observe how the adaptive memory module of a trained agent impacts it's policy. We selectively analyze specific levels, focusing on visualizing the count of pondering steps taken per input. This exploration aims to provide valuable insights into the intricate connection between acquired pondering behaviors and the complexity of inputs, contributing to a comprehensive understanding of the dynamics at play.

We highlight certain observed behaviors from a trained AMM agent with 9 maximum possible repetitions (Figures 5, 7). More examples can be found with different repetition lengths in the appendix. The following behaviors are observed:

**Increasing certainty**: We observe that a trained AMM agent will sometimes choose to ponder for longer with the effect of steadily increasing the probability of it's initial plan (the chosen action with the highest probability). This is analogous to the case of thinking longer to confirm an initial plan. This is the most commonly

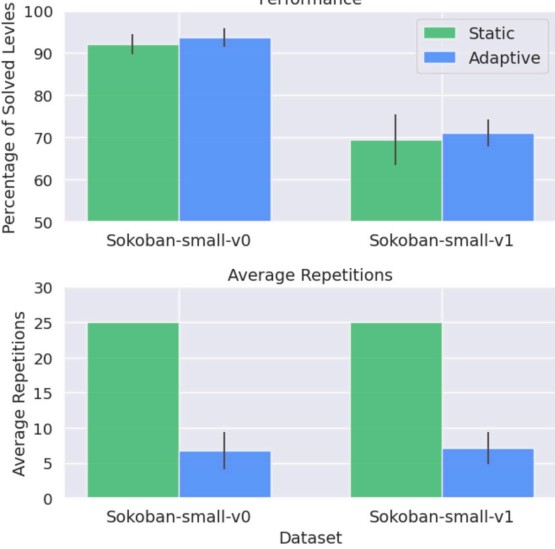

Figure 4: Mean fraction solve at different steps of training, comparing resetting vs reusing hidden state.

observed use case for pondering by our trained model. We also note that the policy's entropy fol-

lows a downward trend, reinforcing the hypothesis that the agent ponders for longer to decrease it's uncertainty. An example of this behavior has been shown in Figure 5.

**Exploring alternatives or Deliberation**: This occurs when throughout pondering the agent's policy seriously considers one or more actions (temporary increase in probability during deliberation), but the final decision remains the same as the initial decision. Figure 5, shows one such example. This type of behaviour is easily compared to forms of exploration, where the policy's entropy is adjusted for exploration. In fact, we can observe this fluctuation in the agent's policy entropy, which shows a temporary increase when the agent begins to more seriously consider an alternative and then begins to fall again.

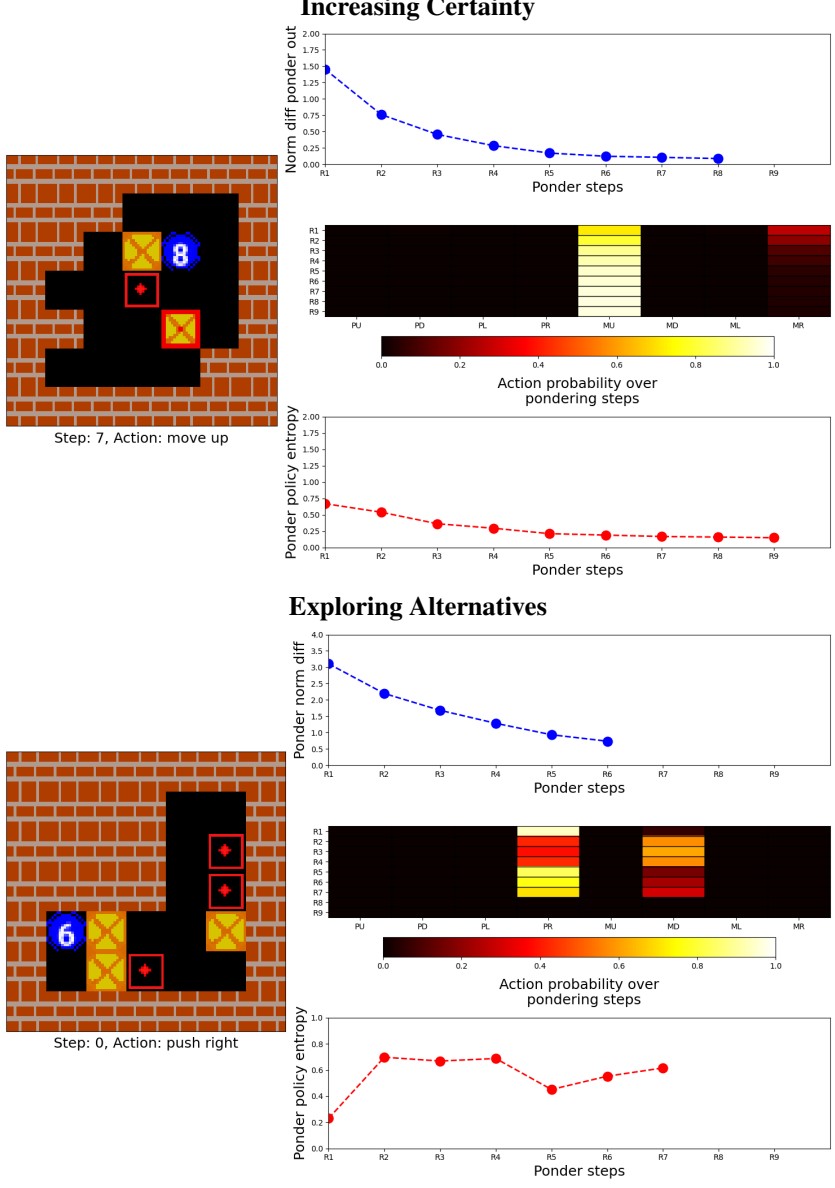

Figure 5: Visualization of a trained AMM agent. Top: *Increasing Certainty*. Bottom: *Exploring Alternatives*. Each image is divided into left and right. Left: the game state with the number of steps pondered at that state displayed on the agent (blue); the current step and the resulting action chosen after pondering is displayed below the image. Right top: the difference in norm of the output between pondering steps. Right middle: the action probability distribution across actions and pondering steps. Right bottom: the entropy of the policy across ponder steps.

**Reconsideration**: Lastly, the phenomenon of reconsideration manifests when an agent's initial and final choices of action differ across the sequence of pondering steps (Figure 6; appendix for more examples). This behavior is characterized by fluctuating levels of certainty after the initial pondering step, followed by a gradual decrement in the probability assigned to the initial action. Subsequently, one or more alternative actions observe a surge in their probabilities, culminating in the selection of a single action among the candidates.

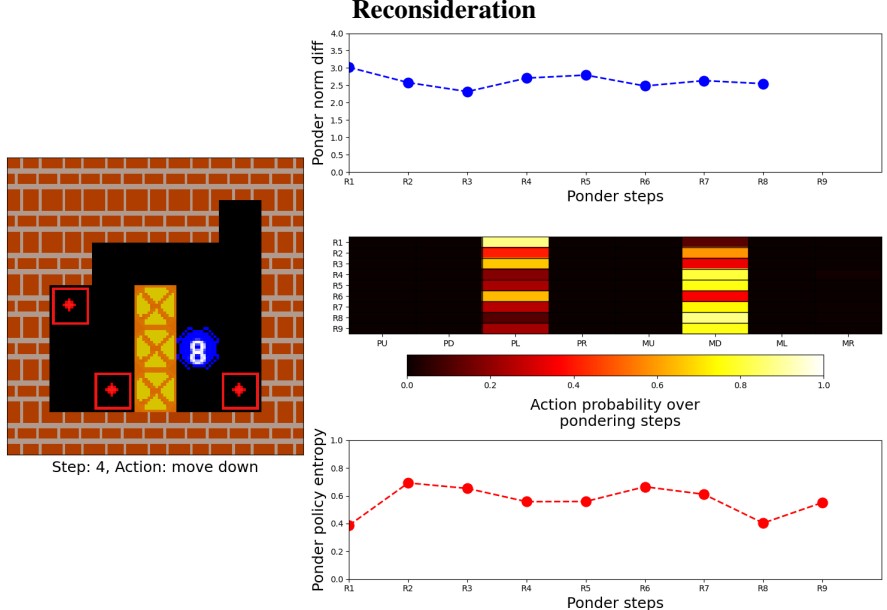

Figure 6: Visualization of a trained AMM agent. Left: the game state with the number of steps pondered at that state displayed on the agent (blue). Right top: the difference in norm of the output between pondering steps. Right middle: the action probability distribution across actions and pondering steps. Right bottom: the entropy of the policy across ponder steps.

## 5    CONCLUSION

The series of experiments detailed in the preceding section highlight the compelling potential of adaptive computation in bolstering planning capabilities and efficiency. By learning to dynamically adjust the duration of deliberation according to the input, agents using AMM, not only improve the inference efficiency on easier inputs, but also improve generalization on more challenging puzzles by deliberating longer. The integration of memory and reusability mechanisms further empowers our models to reuse planning information across time steps, plan deeper and generalize better in sequential decision making domains. Through visualizations of the AMM, we identified learned behaviors (increasing certainty, exploration of alternatives, and reconsideration) that resemble similar strategies leveraged by humans for complex reasoning tasks. Moreover, a fundamental question arises: Can we observe the emergence of additional, potentially novel planning-oriented behaviors by scaling the models, and expanding the training dataset?

Our study, however, has limitations. Our experiments were confined to a specific set of environments necessitating wider explorations. It would be useful to incorporate adaptive computation into other popular architectures like transformers and study the impact of architecture choice.

We hope that the insights gleaned from our results will serve as a catalyst for more profound and expansive analyses of adaptive computation's potential. Particularly pertinent is the role that efficient adaptive computation could play in the training and deployment of progressively larger models in complex tasks requiring reasoning. This could potentially pave the way for more resource-efficient AI systems that continue to deliver state of the art performance through refined adaptive computation strategies.

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

# A APPENDIX

## A.1 ENVIRONMENT AND DATASET DETAILS

Data details:

| Dataset Name | Number of Samples | Description |
|---|---|---|
| Sokoban-small-v0-train | 2.1M | Training data; 7x7 grid, 2 boxes |
| Sokoban-small-v0-validation | 1000 | Validation in-distribution; 7x7 grid, 2 boxes |
| Sokoban-small-v1-validation | 1000 | Validation out-of-distribution; 7x7 grid, 3 boxes |

Table 2: Summary of the datasets.

We decided to omit the inclusion of the no-op action which has the effect that the agent performs no operation, while the environment may or may not change (if there are no other moving objects, it will remain the same input). Also referred to as "stay" actions, many variations on the same idea have been implemented with the goal being to include stochasticity which may help learn a better performing policy, with aperiodic sequence of actions (Hessel et al., 2017). It is important to note that the internal state of the memory module will not be reinitialized. This can be seen as a way to grant the ability to ponder for any agent by allowing it to do nothing and watch it's environment, similar to the no-op action. The reasoning for this is that the AMM already includes the possibility of adaptive computation for an input, and does not entangle the concept of thinking and action by including no-op in the space of possible actions.

## A.2 HYPERPARAMETERS

| Hyperparameter | Value |
|---|---|
| Memory Architecture | DRC(X,Y) |
| Learning Rate | 0.01 |
| Batch Size (number of episodes) | 64 |
| Epochs | 1 |
| Optimizer | AdamW |
| Weight Decay | $1 \times 10^{-4}$ |
| Hidden Units | 128 |
| Dropout Rate | 0.1 |
| Activation Function | ReLU |
| Learning Rate Schedule | Linear warmup for $6.4 \times 10^4$ steps then cosine annealing |
| Gradient Clipping | 1.0 |
| Embedding Size | 50 |
| Recurrent Units | 64 |
| Initialization | He Normal |
| Regularization | None |
| Total Number of Parameters | |

Table 3: Hyperparameters for static models.

| Hyperparameter | Value |
|---|---|
| Memory Architecture | DRC(X,Y) |
| Learning Rate | 0.01 |
| Batch Size (number of episodes) | 64 |
| Epochs | 1 |
| Optimizer | AdamW |
| Weight Decay | $1 \times 10^{-4}$ |
| Hidden Units | 128 |
| Dropout Rate | 0.1 |
| Activation Function | ReLU |
| Learning Rate Schedule | Linear warmup for $6.4 \times 10^4$ steps then cosine annealing |
| Gradient Clipping | 1.0 |
| Embedding Size | 50 |
| Recurrent Units | 64 |
| Initialization | He Normal |
| Regularization | Kullback-Liebler for $\lambda$ |
| Total Number of Parameters | |

Table 4: Hyperparameters for adaptive models.

## A.3 RESULTS

| Static Models | | | | | | |
|---|---|---|---|---|---|---|
| Number of Repetitions | Sokoban-small-v0 | | | Sokoban-small-v1 | | |
| | $6.4 \times 10^4$ | $1.28 \times 10^6$ | $2.4 \times 10^6$ | $6.4 \times 10^4$ | $1.28 \times 10^6$ | $2.4 \times 10^6$ |
| 1 | 60.69 (5.7) | 86.1 (2.8) | 90.01 (3.1) | 32.00 (4.0) | 57.76 (6.9) | 62.3 (5.2) |
| 3 | 67.59 (3.0) | 90.06 (2.4) | **92.37** (2.9) | 35.30 (2.5) | 62.82 (4.6) | 66.19 (4.3) |
| 9 | 72.75 (5.0) | **91.77** (2.3) | 92.36 (2.7) | 42.71 (4.8) | 63.97 (7.0) | 67.30 (5.6) |
| 25 | **76.30** (4.5) | 91.47 (3.4) | 92.10 (2.4) | **47.97** (6.5) | **65.83** (5.8) | **69.39** (6.0) |
| Adaptive Models | | | | | | |
| Number of Repetitions | Sokoban-small-v0 | | | Sokoban-small-v1 | | |
| | $6.4 \times 10^4$ | $1.28 \times 10^6$ | $2.4 \times 10^6$ | $6.4 \times 10^4$ | $1.28 \times 10^6$ | $2.4 \times 10^6$ |
| 3 | 63.86 (1.2) | 88.67 (1.2) | 92.07 (1.1) | 28.51 (1.0) | 61.90 (1.4) | 65.74 (0.9) |
| 9 | **76.56** (2.2) | **91.07** (2.3) | **93.70** (2.2) | **45.05** (2.6) | **68.22** (1.9) | **71.03** (3.2) |
| 25 | 61.89 (10.8) | 90.33 (3.6) | 92.86 (2.0) | 27.20 (7.1) | 65.21 (4.4) | 68.89 (3.0) |

Table 5: Percentage of mean fraction solved at different steps of training for static versus dynamic models. Both dataset tested are from a subset of unseen data from either in-distribution (Sokoban-small-v0), or out-of-distribution (Sokoban-small-v1).

| Static Models | | | | | | |
|---|---|---|---|---|---|---|
| Repetitions | Sokoban-small-v0 | | | Sokoban-small-v1 | | |
| | Mean | Std | Range | Mean | Std | Range |
| 1 | 12.14 | 6.91 | 61.00 | 17.65 | 9.76 | 89.00 |
| 3 | 11.99 | 6.39 | 37.00 | 17.84 | 9.57 | 77.00 |
| 9 | 11.95 | 6.30 | 39.00 | 16.84 | 8.29 | 46.00 |
| 25 | 12.16 | 6.75 | 44.00 | 16.89 | 8.46 | 62.00 |
| Adaptive Models | | | | | | |
| Repetitions | Sokoban-small-v0 | | | Sokoban-small-v1 | | |
| | Mean | Std | Range | Mean | Std | Range |
| 3 | 12.15 | 7.10 | 86.00 | 16.46 | 8.08 | 57.00 |
| 9 | 11.93 | 6.38 | 41.00 | 17.18 | 9.09 | 67.00 |
| 25 | 11.78 | 6.46 | 61.00 | 16.18 | 7.85 | 75.00 |

Table 6: Length of solved episodes statistics for both sets of validation (in-distribution and out-of-distribution). This represents the number of steps taken to complete a level.

| Repetitions | Sokoban-small-v0 | | | Sokoban-small-v1 | | |
|---|---|---|---|---|---|---|
| | Mean | Std | Range | Mean | Std | Range |
| 3 | 1.12 | 0.93 | 2.0 0 | 1.19 | 0.94 | 2.00 |
| 9 | 6.76 | 2.66 | 8.00 | 7.14 | 2.31 | 8.00 |
| 25 | 22.50 | 5.44 | 24.00 | 22.7 | 5.13 | 24.00 |

Table 7: Pondering statistics across validation datasets. Shown is the result of taking the mean, standard deviation, and range of the number of ponder steps taken by a trained AMM over 1000 levels for each environment.

## A.4 DRC ARCHITECTURE

The Deep Repeated ConvLSTM (DRC) architecture (Guez et al., 2019) relies on the ConvLSTM (Shi et al., 2015) module. The high-level idea behind ConvLSTM is to maintain the spatial characteristic of hidden/cell states, by treating those internal states as images. ConvLSTM follows similar computational rules as the standard LSTM, with three-dimensional hidden/cell states and convolutional operations. The DRC architecture, as illustrated in Figure 1, stacks $D$ ConvLSTM modules and repeats the rollout $N$ times at each timestep $t$.

We denote $f_\theta : \mathcal{H} \times \mathcal{X} \rightarrow \mathcal{H}$ as the function that computes the next state $h' \in \mathcal{H}$ given the current state $h \in \mathcal{H}$ and input tensor $x \in \mathcal{X}$ as $h' = f_\theta(h, x)$. The state $h$ refers to the concatenated cell states $c_d$ and hidden states $g_d$ across all stack depth $d \in \{1, \ldots, D\}$, i.e., $h = (c_1, \ldots, c_D, g_1, \ldots, g_D)$, and $\theta = (\theta_1, \ldots, \theta_D)$ is the parameter of all ConvLSTM modules. Let $h_{t-1}$ denote the state at the previous timestep $t - 1$, at the current timestep $t$, the new state $h_t$ is obtained by repeatedly apply $f_\theta$:

$$h_t = s_\theta(h_{t-1}, x_t) = \underbrace{f_\theta(f_\theta(\ldots f_\theta(h_{t-1}, x_t), \ldots x_t), x_t)}_{N \text{ times}} \tag{4}$$

Unlike the original DRC introduced by Guez et al. (2019), in our work DRC outputs the hidden state of the deepest ConvLSTM module $g_D^{(n)}$ at each repetition $n \in \{1, \ldots, N\}$. Based on the hidden state $g_D^{(n)}$ we compute the policy $\pi^{(n)}$ and halting probability $\lambda^{(n)}$. We also set the depth $D = 1$ in our experiments. The rest of the architecture remains the same as the original work.

## A.5 ADDITIONAL VISUALIZATIONS

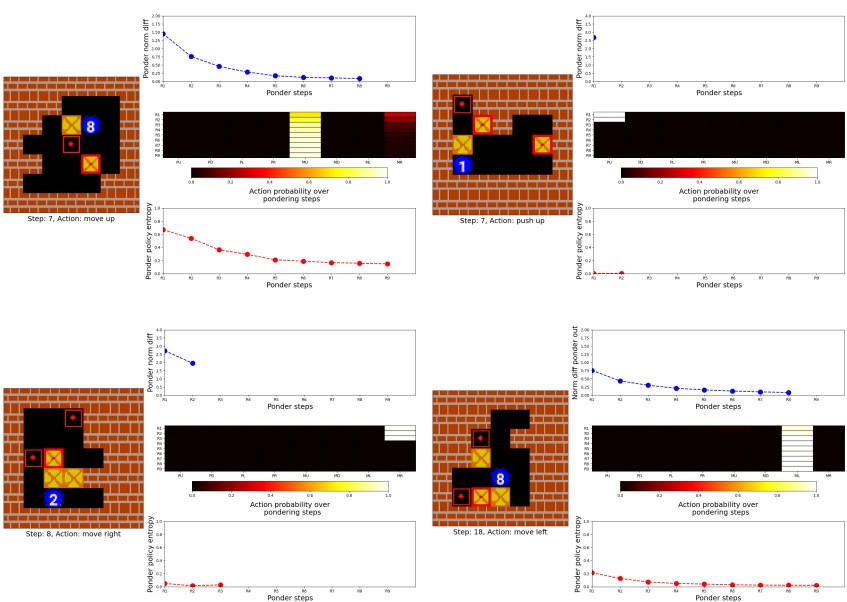

Figure 7: **Increasing certainty**

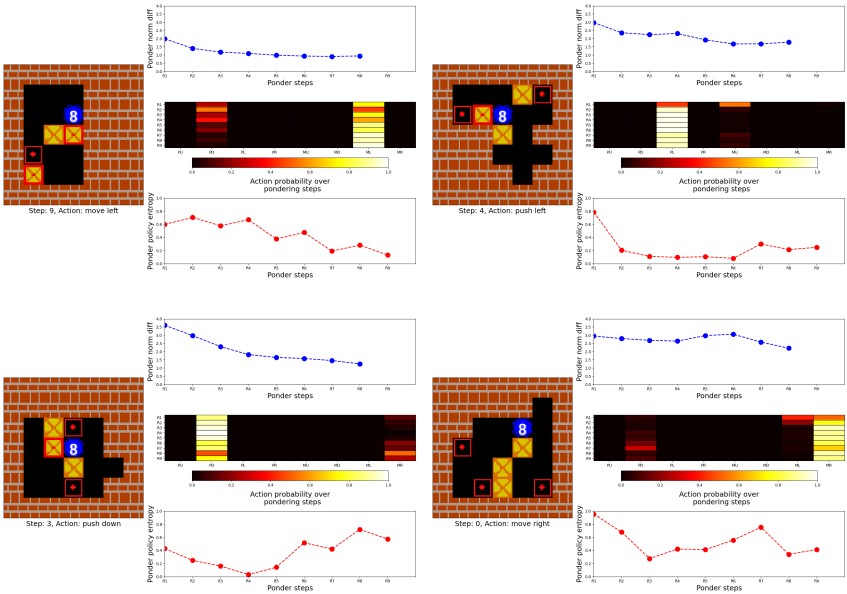

Figure 8: **Exploring alternatives**

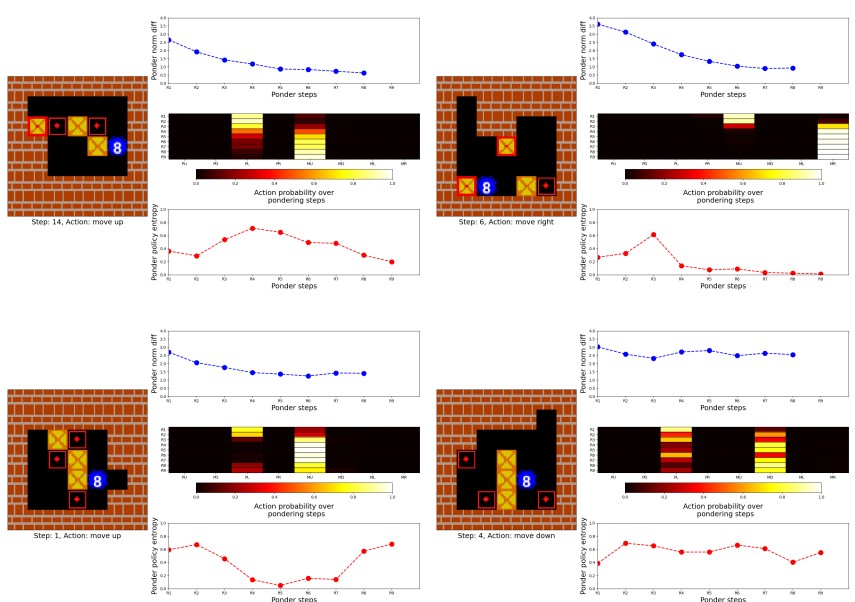

Figure 9: **Reconsideration**

