# OpenReview forum: "Adaptive Memory Module for Sequential Planning and Reasoning"
_ICLR.cc/2024/Conference — ICLR 2024 Conference Withdrawn Submission_

### Official Review · Reviewer_gpnE · 2023-10-29

**Soundness:** 1 poor
**Presentation:** 1 poor
**Contribution:** 1 poor
**Rating:** 1
**Confidence:** 4

**Summary:**

This paper introduces an adaptive memory module based on the combination of the DRC (Deep Repeated ConvLSTM) architecture and the PonderNet algorithm. The framework is then applied in a simple RL environment (Sokoban Puzzle) for behavior cloning tasks.

**Strengths:**

- The paper rejuvenates interest in the PonderNet algorithm, underscoring its potential for future research.
- Empirical evidence demonstrates the agent's adeptness in resource allocation contingent on data intricacy, offering valuable insights for RL tasks, notably in curriculum design.

**Weaknesses:**

- The narrative is ambiguous, detracting from a clear understanding.
- The framework mirrors pre-existing models, questioning its novelty.
- The study lacks both theoretical depth and comprehensive experimentation. Presented tests are elementary and dated, with no discernible real-world significance.

**Questions:**

I suggest the authors to consider applying the adaptive memory module in large language models with prompt engineering. How to design a practical algorithm for an LLM-based agent to save and retrieve memories adaptively according to the task difficulty and budget constraints? This would be a way more impactful problem.

---

### Official Review · Reviewer_KTzw · 2023-10-30

**Soundness:** 1 poor
**Presentation:** 1 poor
**Contribution:** 1 poor
**Rating:** 1
**Confidence:** 4

**Summary:**

This paper studies the impact of external memory and adaptative computation of a learning agent in the Sokoban Puzzle environment

**Strengths:**

Decision-making and planning is a relevant issue for the ICLR community, authors also do multiple ablations of the proposed architecture,

**Weaknesses:**

I am afraid that at its current state the paper is far from ready for publication in a conference. The paper is plagued of fundamental weaknesses, e.g.,  figures lack confidence intervals in the plots of the experimental section, the empirical analysis is done only in  one environment, there is not theoretical background section, neither authors do a theoretical grounding of the proposed architecture, there is a lack of comparison with existing methods papers that do analysis of external and adaptative memories in learning agents, i.e. lack of literature review, most clames aren't correctly supported, presentation and writing needs a lot of extra work and review. There are also some statements that are also fundamentally wrong (examples below)

**Questions:**

Some examples about writing and no supporting correctly the claims in the paper include:
* First paragraph "reasoning in sequential decision making domains is characterized by an interaction between
the past and the future, where information and decisions from previous steps significantly inform
the present decisions while these decisions, in turn, influence future inputs and observations" This statement is not true, the dominant technique, RL, is based on the Markovian assumption about the past.

* Second paragraph  "simply scaling up model size, and dataset size can result in emergent capabilities like reasoning" while scaling has brought emergent capabilities, to say that LLMs have reasoning it is something controversial at the very least. Also ".Factors contributing to these limitations include low-quality step-by-step reasoning
data available on the internet and difficulty in collecting high-quality reasoning data for complex reasoning problems." Here lacks at least a reference

* Third paragraph, model-based RL do not construct "mental" simulations. Also there, you start referring to planning routines without specifiying what you mean

* Paragraphs 4 and 5 seem to be out of place, I would suggest to use that part to build intuition about the architecture. It is not a good approach to mention for the first time that the paper is proposing an architecture at the end of the second page.

* The theorethical foundation need further detail, intuition and justification. For instance  you say that there is a recurrent memory module and that you use DRC, but then you also say that DRC can be substituted by MLPs or attention networks which are not recurrent, this cast doubts over the soundness of your work.

* Also there are plenty of statements such as "The internal memory is not reset after each step in the
environment, enabling the agent to reuse past planning information. This allows the agent to plan
deeper into the future and generalize to more complex reasoning problems."  that feel like this work is reselling things that are long stablished in the literature, again without going into the details on how is this different and why are you introducing such changes.

---

### Official Review · Reviewer_Us5J · 2023-10-31

**Soundness:** 3 good
**Presentation:** 3 good
**Contribution:** 1 poor
**Rating:** 3
**Confidence:** 3

**Summary:**

The paper presents an architecture called Adaptive Memory Module. It is designed to implicitly learn skills such as reasoning, reusing information, and adapting the computing time to the complexity of the task. The model is tested experimentally in the Sokoban environment.

**Strengths:**

The paper is well-structured and easy to follow. The model description is clear and concise. The experimental evaluation is comprehensive.

**Weaknesses:**

My main concern is that there is little takeaway from this work. The proposed architecture is a combination of existing approaches. Mainly [1], cited in the paper, which I see is quite closely followed. Additionally, there is a large body of research on conditional computation, see e.g. [2], [3], [4] to name a few. This relation is completely omitted in the paper (as far as I see), which unfortunately withdraws much of the novelty. This itself is not destructive, since a combination of existing approaches can be novel. However, I don't think that in this case it is justified. The model is validated in a single environment, one that is not very challenging. If you can indeed show that the presented architecture is capable of complex reasoning, I'd be satisfied. However, this is impossible in this simple environment. Although the experimental analysis is comprehensive, barely any conclusions can be made.

The notation of _repetitions_ should be introduced, because it's not obvious whether you mean N, D, or any other dimension.

The impact of repetitions seems very low. Even in case of the harder instances, the difference between 1 and 25 lies within a 9% interval. Thus, I don't see strong benefits of using repetitions at all.

Figures 3 and 4 seem to be swapped, according to the references in the text.

When testing adaptivity in Section 3.4, I understand that the adaptive variant uses at most 9 layers. However, as you observed earlier, the models with fewer repetitions shouldn't in general outperform the models with more repetitions. Why then it is better than the static model with 25 repetitions? It seems like the outcome of variance. Anyway, I believe that the adaptive model should be compared with an equally deep model, so 9 layers. I see that in this case the computational advantage would be rather small (6.76 to 9)

I like the visualizations in Section 4. However, I think that the conclusions are rather trivial in that the initial estimate can be either kept, changed, or reconsidered. These are basically all the options. I admit that it can be seen from the perspective of human-like skills (confirmation, exploration, reconsideration), but sadly there are no further takeaways from this experiments.

In general, I'm not convinced that the presented architecture brings us closer to achieving the reasoning skills. It seems like simply a wider and deeper recurrent policy with early exits. All the concepts are already widely known in RL and in various forms and combinations.

[1] Adaptive Computation Time for Recurrent Neural Networks

[2] Shallow-Deep Networks: Understanding and Mitigating Network Overthinking

[3] Why should we add early exits to neural networks?

[4] Zero Time Waste: Recycling Predictions in Early Exit Neural Networks

**Questions:**

Please specify the differences between your work and [1]. Also please, compare your approach with the line of work on early exits.

How large are the datasets that you use for training? How long are the trainings in terms of wall time?

Why the 1-9 adaptive model is better than the static model with 25 repetitions? It seems like the outcome of variance, can you explain it?

---

### Official Review · Reviewer_6Gb2 · 2023-11-02

**Soundness:** 1 poor
**Presentation:** 1 poor
**Contribution:** 1 poor
**Rating:** 1
**Confidence:** 4

**Summary:**

The paper aims to analyze the effect of memory modules in decision-making problems in the puzzle-solving task of Sokoban. They also analyze the number of "pondering" steps carried out by an adaptive agent in response to the complexity of a state.

**Strengths:**

- The motivation for adaptive computation makes sense and can be an important research direction.
- The empirical study has several experiments, including qualitative studies. But it is hard to understand what the experiments are trying to target or prove.

**Weaknesses:**

## Writing and Presentation
- The paper is written in a manner that makes it really hard to follow and understand what is the problem that is addressed, what are the novel insights from this paper and what is taken from prior work, and what are the hypotheses that must be tested in experiments. It is written in a very linear manner that tells what the authors did, but there is no top-down story or description that a new reader can follow and understand. Naturally, it was difficult to review this paper.
- What is pondering? Introduction just mentions the "complexity-pondering" relationship without every explaining what pondering means technically.

## Unclear contributions
The paper states: "In this study we aim to explore how agents can leverage adaptive computation, memory, and an end-to-end learning framework to meta-learn flexible planning strategies and enhance decision time efficiency while retaining the ability for further deliberation to solve more complex problems."
- This is a very complicated statement that comes out of nowhere. The prior work context provided before this line does not explain what is going on here. For instance, what is the relevance of memory and meta-learning?
- There is no central problem introduced in the paper, and thus it is unclear what is the novelty of this work and what are the baselines that should be compared against.
- The methodology section is just introduced without knowing what is the problem this method is solving. If the problem is to "analyze memory mechanisms", then why this particular method architecture? Also, is this method architecture novel? What was taken from prior work and what does this work add?
	+ What are the technical contributions of the proposed method?
	+ Is everything following the prior work of PonderNet? If not, then what is new? There needs to be a very clear distinction between what prior work defines and what this work introduces as a novel solution.

## Experiments
- The experiment section just lists what the authors do, without any flow of what is the problem and hypothesis being tested. Very hard to follow what is going on in the different environment sections.
- The evaluation is also severely limited to only Sokoban puzzles.

**Questions:**

Many questions are asked above in the Weaknesses section. Most importantly,
- What is the problem being solved for which the insight of the adaptive memory module is important?